# Effect of aerobic exercise on amyloid accumulation in preclinical Alzheimer's: A 1-year randomized controlled trial

Eric D. Vidoni[1], Jill K. Morris[1], Amber Watts[2], Mark Perry[3], Jon Clutton[1], Angela Van Sciver[1], Ashwini S. Kamat[1], Jonathan Mahnken[1,4], Suzanne L. Hunt[1,4], Ryan Townley[1], Robyn Honea[1], Ashley R. Shaw[1], David K. Johnson[5], James Vacek[6], Jeffrey M. Burns[1]*

1 University of Kansas Alzheimer's Disease Center, Fairway, KS, United States of America, 2 Department of Psychology, University of Kansas, Lawrence, KS, United States of America, 3 Department of Radiology, University of Kansas Health System, Kansas City, KS, United States of America, 4 Department of Biostatistics and Data Science, University of Kansas Medical Center, Kansas City, KS, United States of America, 5 Department of Neurology, University of California–Davis, Sacramento, CA, United States of America, 6 Department of Cardiovascular Medicine, University of Kansas Health System, Kansas City, KS, United States of America

* jburns2@kumc.edu

**Data Availability Statement:** All files are available from the Harvard Dataverse database (Vidoni, Eric, 2020, "Alzheimer's Prevention Through Exercise (APEx) - NCT02000583", https://doi.org/10.7910/

## Abstract

### Background

Our goal was to investigate the role of physical exercise to protect brain health as we age, including the potential to mitigate Alzheimer's-related pathology. We assessed the effect of 52 weeks of a supervised aerobic exercise program on amyloid accumulation, cognitive performance, and brain volume in cognitively normal older adults with elevated and sub-threshold levels of cerebral amyloid as measured by amyloid PET imaging.

### Methods and findings

This 52-week randomized controlled trial compared the effects of 150 minutes per week of aerobic exercise vs. education control intervention. A total of 117 underactive older adults (mean age 72.9 [7.7]) without evidence of cognitive impairment, with elevated (n = 79) or subthreshold (n = 38) levels of cerebral amyloid were randomized, and 110 participants completed the study. Exercise was conducted with supervision and monitoring by trained exercise specialists. We conducted 18F-AV45 PET imaging of cerebral amyloid and anatomical MRI for whole brain and hippocampal volume at baseline and Week 52 follow-up to index brain health. Neuropsychological tests were conducted at baseline, Week 26, and Week 52 to assess executive function, verbal memory, and visuospatial cognitive domains. Cardiorespiratory fitness testing was performed at baseline and Week 52 to assess response to exercise. The aerobic exercise group significantly improved cardiorespiratory fitness (11% vs. 1% in the control group) but there were no differences in change measures of amyloid, brain volume, or cognitive performance compared to control.

DVN/B9I1F8, Harvard Dataverse, V1, UNF:6:
pgNe0pp64djpi7AAdwhBiw== [fileUNF]).

**Funding:** This work was supported by the National
Institutes of Health R01 AG043962 (JMB); K99
AG050490 (JKM) and gifts from Frank and
Evangeline Thompson (JMB), The Ann and Gary
Dickinson Family Charitable Foundation, John and
Marny Sherman, and Brad and Libby Bergman.
Institutional infrastructure support for testing was
provided in part by UL1 TR000001 (RJB) and P30
AG035982 (RHS JMB). Lilly Pharmaceuticals
provided a grant to support F18-AV45 doses and
partial scan costs (JMB). The content is solely the
responsibility of the authors and does not
necessarily represent the official views of the
National Institutes of Health. The funders had no
role in study design, data collection and analysis,
decision to publish, or preparation of the
manuscript.

**Competing interests:** This work was supported by
the National Institutes of Health R01 AG043962
(JMB); K99 AG050490 (JKM) and gifts from Frank
and Evangeline Thompson (JMB), The Ann and
Gary Dickinson Family Charitable Foundation, John
and Marny Sherman, and Brad and Libby
Bergman. Institutional infrastructure support for
testing was provided in part by UL1 TR000001
(RJB) and P30 AG035982 (RHS JMB). Lilly
Pharmaceuticals provided a grant to support F18-
AV45 doses and partial scan costs (JMB). The
content is solely the responsibility of the authors
and does not necessarily represent the official
views of the National Institutes of Health. The
funders had no role in study design, data collection
and analysis, decision to publish, or preparation of
the manuscript. This does not alter our adherence
to PLOS ONE policies on sharing data and
materials.

## Conclusions

Aerobic exercise was not associated with reduced amyloid accumulation in cognitively normal older adults with cerebral amyloid. In spite of strong systemic cardiorespiratory effects of the intervention, the observed lack of cognitive or brain structure benefits suggests brain benefits of exercise reported in other studies are likely to be related to non-amyloid effects.

## Trial registration

NCT02000583; ClinicalTrials.gov.

## Introduction

There is increasing interest in the role of exercise in the prevention and treatment of Alzheimer's disease and related cognitive disorders given the growth of the older adult population. Though not all studies agree [1], accumulating evidence suggests that aerobic exercise may protect against cognitive decline and dementia [2–5]. Ongoing work will provide more definitive evidence regarding the cognitive benefits of exercise [6], but aerobic exercise remains among the most promising and cost-effective strategies for delaying or preventing cognitive decline and dementia [7,8].

A wealth of data indicate exercise positively impacts brain health. Higher levels of aerobic fitness are associated with age-related improvements or attenuated decline in brain volume and cognition at both cross-section and over time [3,9–13]. In randomized controlled trials, aerobic exercise promotes brain plasticity, attenuate hippocampal atrophy, or even promotes hippocampal volume increases while improving spatial memory [3,5,14,15].

The effect of aerobic exercise on the pathophysiological markers of AD, beta-amyloid and tau, have been less well explored. Animal studies indicate exercise may reduce amyloid burden and modify AD pathophysiology through direct effects on amyloid precursor protein metabolism [16–18] and indirect effects on neurotrophic factors, neuroinflammation, and oxidative stress [16,17,19–21]. Exercise-induced reductions of amyloid also appear to mediate improvements in cognitive functioning in animals [22–25]. Studies in humans assessing the effect of physical activity on AD pathophysiology are limited. Cross-sectional, observational studies in humans have found that greater amounts of self-reported physical activity (i.e., volitional behavior that is part of daily function) is associated with evidence of lower cerebral amyloid levels among cognitively normal adults [26–32], and those at high genetic risk for AD [26,33–35]. It remains unclear whether the lifestyle behaviors causally influence cerebral amyloid, or vice versa, and whether introducing more physical activity through planned exercise can causally mitigate amyloid pathology.

The advent of amyloid imaging creates an opportunity for identifying individuals in the presumptive pre-symptomatic phase of AD, when interventions may have the greatest impact [36]. Approximately 30% of cognitively normal older adults have asymptomatic cerebral amyloidosis and thus meet the NIA and Alzheimer's Association research criteria for "preclinical AD", defined as having a cerebral to cerebellar amyloid ratio above a certain, method-dependent threshold. The concept of preclinical AD posits that cerebral amyloid deposition in cognitively normal adults represents a pre-symptomatic stage of AD and individuals with preclinical AD currently represent the earliest feasible stage for trials of AD prevention. Individuals with subthreshold levels of cerebral amyloid (individuals with non-elevated amyloid PET but with quantitative measures near the threshold for being elevated) may be more likely to accumulate

clinically significant levels of amyloid and have memory decline [37], suggesting they are good candidates for prevention studies [38].

Our study examined the effects of a 52-week aerobic exercise program on AD pathophysiology (amyloid burden), associated "downstream" neurodegeneration (whole brain and hippocampal volume change) and cognitive decline in cognitively normal individuals with either preclinical AD or with subthreshold levels of cerebral amyloid. We hypothesized that 52 weeks of aerobic exercise would be associated with reduced amyloid accumulation, reduced hippocampal atrophy, and improved performance on a cognitive test battery.

## Materials and methods

### Study design

The Alzheimer's Prevention through Exercise study (APEx: ClinicalTrials.gov, NCT02000583; trial active between 11/1/2013–11/6/2019) was a 52-week study of aerobic exercise in individuals 65 years and older without cognitive impairment. Based on public health recommendations and our prior work [4,39], we randomized individuals to either 150 minutes per week of supported moderate intensity aerobic exercise or standard of care education control in a 2:1 ratio. The unbalanced design was intended to maximize recruitment and retention with minimal impact in power. Cerebral amyloid load, neurodegeneration, cognition, and cardiorespiratory fitness were measured at baseline and post-intervention. Cognition was also measured at the midpoint of the study. The University of Kansas Medical Center Human Subjects committee approved the protocol (HSC#13376) and written informed consent was obtained from all participants.

### Participants

Participants were recruited as a convenience sample of volunteers through print and online advertising, community talks, and existing databases of individuals willing to be in research studies [40]. Enrollment occurred between March 1, 2014 and October 31, 2018. Interested individuals first underwent a telephone screen of medical history for key inclusion and exclusion criteria including: age of 65 years and older, sedentary or underactive as defined by the Telephone Assessment of Physical Activity [41], on stable medications for at least 30 days, willingness to conduct prescribed exercise (or not) for 52 weeks at a community fitness center, and willingness to undergo an 18F-AV45 PET scan for cerebral amyloid load and learn their individual result (elevated vs non-elevated). Amyloid status was disclosed to all participants regardless of screening status [42]. In-person screening included a clinical assessment by clinician of the University of Kansas Alzheimer's Disease Center including a Clinical Dementia Rating and Uniform Data Set neuropsychiatric battery [43,44]. Participants could not be insulin-dependent, have significant hearing or vision problems, clinically evident stroke, cancer in the previous 5 years (except for localized skin or cervical carcinomas or prostate cancer), uncontrolled hypertension, or have had recent history (<2 years) of major cardiorespiratory, musculoskeletal or neuropsychiatric impairment, and had to be able to complete graded maximal exercise testing with a respiratory exchange ratio $> = 1.0$.

We enrolled only those participants who met criteria for elevated cerebral amyloid (see below) as previously described [42,45], until March 2016 when we revised the protocol to allow individuals with subthreshold amyloid levels (cerebral-to-cerebellar standard uptake value ratio (SUVR) threshold $> 1.0$). This was motivated by recruitment challenges for the preclinical AD group and new evidence that this group accumulates amyloid and is more likely to have associated memory decline [37] and thus may represent an excellent target for early prevention studies.

## Amyloid screening

Florbetapir PET scans were obtained approximately 50 minutes after administration of intra-venous florbetapir 18F-AV45 (370 MBq) on a GE Discovery ST-16 PET/CT scanner. Two PET brain frames of five minutes in duration were acquired continuously, summed, and attenuation corrected. To determine amyloid status three experienced raters interpreted all images independently and without reference to any clinical information, as previously described [45]. Raters followed a process that combined both visual and quantitative information to determine status as "elevated" vs "non-elevated." Final status was determined by majority of the three raters [46,47]. Images were viewed and analyzed using the MIMneuro Amyloid Workflow (version 6.8.7, MIM Software Inc., Cleveland, OH, USA), using florbetapir templates as the target for a two-phase registration: first rigid registration, then deformable registration to a common template space. Raters first reviewed raw PET images visually then examined the cerebellum normalized SUVRs in 6 cortical regions (anterior cingulate, posterior cingulate, precuneus, inferior medial frontal, lateral temporal, and superior parietal cortex) and projection maps comparing SUVRs to an atlas of amyloid negative scans [46]. Participants were eligible for the study if they had an elevated scan or (after March 2016) were in the subthreshold range. We defined subthreshold as a mean cortical SUVR for the 6 ROIs > 1.0, which represented the upper half of non-elevated scans (mean cortical SUVR for non-elevated scans [n = 166] 0.99 [0.06 SD]). Enrolled participants were re-scanned after 52 weeks of intervention.

## Allocation

A study statistician constructed an allocation schedule that was applied by study staff after baseline testing was completed. The study statistician used random number generator to generate blocks of nine in a 2:1 ratio to protect against imbalance if recruitment fell short Participants were prospectively assigned to treatment versus control from this schedule using REDCap's randomization module which restricts access and viewing once uploaded.

## Intervention

Participants in the education control group were provided with standard exercise public health information and received a membership to a community exercise facility after completion of the study.

For those randomized to the aerobic exercise group, the intervention was conducted at their nearest study-certified exercise facility under the guidance of certified personal trainers employed by the community exercise facility. They were asked to refrain from changing their regular physical activities other than those prescribed by the study team. Methods for ensuring study protocol compliance and ongoing training refreshers have been published previously [4,48,49]. Personal trainers oversaw prescription for weekly exercise duration and intensity under the direction of the study team. At each session, participants manually recorded the duration of exercise on an exercise study log. Exercise began with a goal of 60 total minutes during Week 1 and increased by approximately 21 min/week until achieving 150 min/week of aerobic exercise. Participants exercised 3–5 days a week, never more than 50 minutes a day to reduce the likelihood of overuse injury. Intensity was prescribed as a target heart rate zone (F4 or FT4, Polar Electro Inc., Lake Success, NY) based on the percentage of heart rate reserve (HRR) as calculated by the Karvonen formula. Beginning at 40–55% of HRR (% of the difference between maximal and resting), target heart rate zones were increased by 10% of HRR every 3 months.

Trainers supervised all exercise sessions for the first 6 weeks of exercise and at least once weekly thereafter. Treadmill walking served as the primary exercise mode but participants

were allowed to use a different aerobic modality if requested to alleviate boredom or accommodate discomfort. No compensation was provided to participants beyond the fees paid to the exercise facility for memberships and trainer time. We have previously demonstrated that our methods using community fitness facilities and trainers can deliver a well-controlled exercise dose with rigor and a high level of adherence, comparable to lab-based methods [4,49].

## Adherence and safety

Trainers asked about changes in health status (adverse events [AE]) at every visit. Study staff inquired about AEs and medication changes during scheduled telephone check-ins every 6 weeks, or during incidental contact at weekly exercise facility visits. An independent safety committee reviewed AEs quarterly. Intent-to-treat analyses were performed on all enrollees (n = 117). We separately assessed individuals who participated in the trial per-protocol (n = 92) by complying with at least 80% of the intervention exercise prescription [4].

## Outcomes

This study sought to provide evidence of a specific effect on AD pathophysiology (i.e., disease-modifying effect) of aerobic exercise on AD-related pathophysiological change in preclinical AD. We specified our primary outcome as mean change from baseline to 52 weeks in 18F-AV45 standard uptake value ratio (SUVR) with secondary outcomes of MRI measures of change in whole brain and hippocampal volume and cognitive performance measures. To assess the physiologic impact of the intervention, we measured the highest achieved oxygen consumption rate (VO$_2$ peak, mL·kg$^{-1}$·min$^{-1}$) during a graded exercise test [4].

Substantial evidence exists demonstrating that aerobic exercise has a preferential effect on cognition, particularly in executive functioning [3,50]. Thus, our cognitive outcome measure of interest was executive function. We also planned to assess key cognitive domains that are associated with asymptomatic cerebral amyloid deposition such as episodic memory and visuospatial function which has been previously associated with aerobic exercise [4]. Raters involved with key outcomes (psychometrician, imaging technicians, exercise physiologists) were blinded to the participant's intervention group (aerobic exercise or control) and had no interaction with participants beyond the testing visits.

**Magnetic Resonance Imaging (MRI) of brain anatomy.** MRI of the brain was performed at baseline and 52-week follow up testing in a Siemens 3.0 Tesla Skyra scanner. We obtained a high-resolution T1 weighted image (MP-RAGE; 1x1x1.2mm voxels; TR = 2300ms, TE = 2.98ms, TI = 900ms, FOV 256mmx256mm, 9˚flip angle) for detailed anatomical assessment. We used the Freesurfer image analysis suite (ver. 5.2 http://surfer.nmr.mgh.harvard.edu/) for volumetric segmentation optimized for longitudinal data [51], extracting hippocampal and total gray matter volume change as measures of neurodegeneration.

**Cognitive test battery.** A trained psychometrist performed a comprehensive cognitive test battery at baseline and again at Week 26 and Week 52, employing validated, alternate versions of tests every other visit. We created composite scores for three cognitive domains (executive function, verbal memory, visuospatial processing) using Confirmatory Factor Analysis (CFA) in MPlus software. We standardized scores to baseline mean and standard deviation, thus scores at Week 26 and Week 52 can be interpreted as a change from baseline. The executive function domain composite score was made up of verbal fluency (the sum of animals and vegetables) [52], Trailmaking Test B [53], Digit Symbol Substitution test [54], and the interference portion of the Stroop test [55]. The verbal memory domain composite score was made up of the immediate and delayed portions of the Logical Memory Test [54], and the sum of free recall trials of the Selective Reminding Test [56]. The visuospatial domain composite score was

made up of scores from Block Design [54], space relations, the paper folding test, hidden pictures, and identical pictures [57]. We included the combined cognitive scores as outcomes in subsequent models. Missing data were accounted for using full information maximum likelihood algorithm. To evaluate model fit, we used Root Mean Squared Error of Approximation (RMSEA), a measure of the discrepancy between predicted and observed model values. Values closer to 0 indicate better fit (preferred values are <0.09). We report a comparative fit index (CFI) that estimates the relative fit of a model compared to an alternative model, in which a CFI >0.90 indicates good fit. Typically, these multiple fit indices are considered together, as opposed to relying on any one indicator.

**Graded maximal exercise test.** We assessed cardiorespiratory fitness at baseline and Week 52 as the highest oxygen consumption attained ($VO_2$ peak) during cardiorespiratory exercise testing on a treadmill to maximal capacity or volitional termination [4].

**Genotyping.** APOE genotype determination Whole blood was collected and stored at -80C until genetic analyses could be conducted. To determine APOE genotype, frozen whole blood was assessed using a Taqman single nucleotide polymorphism (SNP) allelic discrimination assay (ThermoFisher). APOE4, APOE3, and APOE2 alleles were distinguished using Taqman probes to the two APOE-defining SNPs, rs429358 (C_3084793_20) and rs7412 (C_904973_10). The term "APOE4 carrier" was used to describe the presence of 1 or 2 APOE4 alleles.

## Statistical analysis

Descriptive statistics were generated, including means, standard deviations and ranges for continuous measures, and frequencies and relative frequencies for categorical measures. For primary study endpoints with baseline and 12-month follow-up data only, we calculated differences between pre- and post-treatment measures and compared these differences with two-sample t-tests for intent-to-treat analyses. For further assessment among the per protocol population we used covariate adjustment by analyzing these difference scores as a function of the treatment group and other covariates (age, sex, education, and PET amyloid status [elevated vs. subthreshold]) using ordinary least squares regression. For cognition endpoints measured at three time points (baseline, 26-, and 52-weeks), linear mixed models were used. We used a random intercept for subject to account for repeated measures, and treated time as a linear explanatory variable. Unadjusted analyses included treatment group, time, and their interaction, with the interaction test term providing the test for interaction effect using a t-test of that parameter from the model for intent-to-treat results. This approach also allowed for further covariate adjustment for sex, age, education, and PET amyloid status among the per-protocol subgroup.

All statistical methods assessed appropriate model assumptions. For continuous measures, this involved residual analyses to assess normality and variance homogeneity assumptions.

At the time of study design, no previous exercise studies in humans had measured in vivo amyloid. Therefore, we powered our primary outcome from prior investigational compound work. A 78-week study assessing Bapineuzumab in AD reported an effect size of d = 1.98 [58]. Given the differences in our proposed study (preclinical AD sample, lower expected amyloid burden, shorter duration and expectation of a lower exercise effect) we estimated effect size of only 40% as large, (resultant d = 0.79), yielding 93% power to detect this conservative anticipated effect of exercise on amyloid burden. Our initial enrollment goal was 100. Subsequent to including individuals with subthreshold amyloid, we increase our enrollment goal to 120.

Data were captured using REDCap [2] which allowed for secure randomization and role based access to data capture forms. The analysis for this project was generated using SAS software, Version 9.4 for Windows (SAS Institute Inc., Cary, NC, USA).

## Results

### Participants

A total of 1578 individuals were assessed for study eligibility from November 2013 to October 2018. The flow of participants from screening through study completion is shown in Fig 1. Participants (n = 117) were randomized to either the aerobic exercise (n = 78) or control (n = 39) intervention groups.

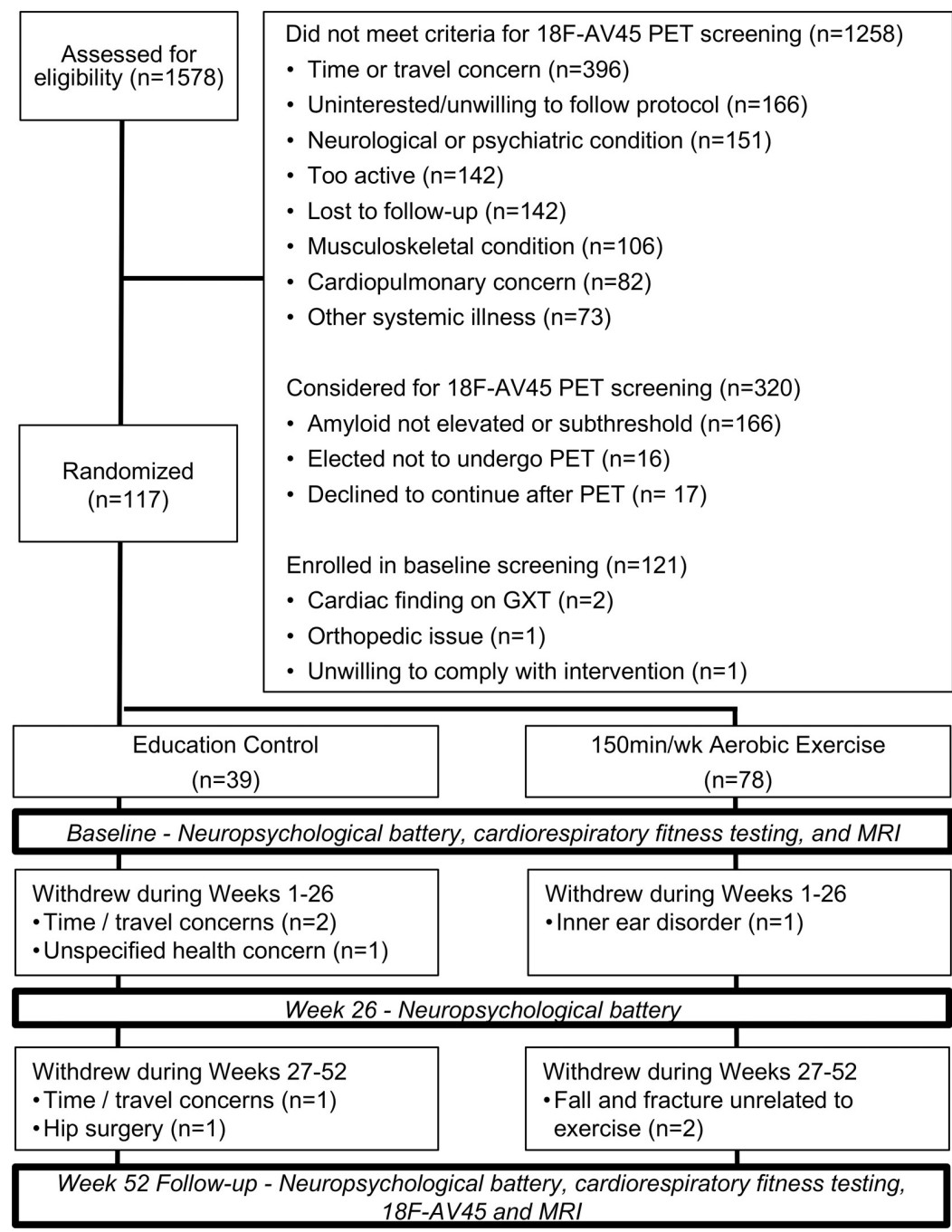

**Fig 1. APEx study CONSORT diagram.**

**Table 1. Enrolled participant demographics.**

| Measure | Education Control (n = 39) | Aerobic Exercise (n = 78) | p-value |
|---|---|---|---|
| Age (y) | 72.2 (5.3) | 71.2 (4.8) | 0.31 |
| Sex, female n(%) | 24 (61.5) | 55 (70.5) | 0.44 |
| Education (y) | 16.2 (2.2) | 16.1 (2.4) | 0.71 |
| APOE ε4 carriers n(%)* | 15 (38.5) | 36 (46.2) | 0.51 |
| Race/Ethnicity: White, not Latino n(%) | 35 (89.7) | 77 (98.7) | 0.08 |
| African American n(%) | 4 (10.3) | 1 (1.3) | |
| Baseline MMSE (mean, range) | 29.1 (26–30) | 29.1 (26–30) | 0.51 |
| Elevated amyloid status: n(%) | 27 (69.2) | 52 (66.7) | 0.94 |
| subthreshold n(%) | 12 (30.8) | 26 (33.3) | |

Mean (standard deviation), unless otherwise noted.

*Four individuals declined genotyping. APOE = Apolipoprotein ε4 genotype; MMSE = Mini-Mental State Exam.

A total of 109 participants (93%: control n = 34, aerobic exercise n = 75) completed the study. There were no significant differences across intervention groups in demographic and baseline characteristics (p>0.05, Table 1).

## Adherence to exercise protocol

The aerobic exercise group completed an average of 84.6% (SD 25.8%) minutes of the pre-scribed exercise dose. The control group did not report weekly exercise. However, the control group remained underactive or sedentary during the intervention as evidence by their self-report of weekly physical activity [59]. The control group reported a -778 calorie (SD 5101) reduction in moderate intensity activity from baseline to Week 52. In contrast, the aerobic exercise group increased moderate intensity physical activity by 1853 calories (SD 5019).

## Outcomes of interest

We provide pre-specified analyses for both the intent-to-treat cohort of 117 enrollees and a per-protocol cohort of 92 individuals who were protocol adherent (as defined as achieving > = 80% of prescribed exercise minutes).

Primary and secondary outcomes are detailed in Table 2. In the intent-to-treat group, there was a strong physiologic effect of aerobic exercise on cardiorespiratory fitness, with the aerobic exercise group increasing $VO_2$ peak by 11% compared to 1% in the control group. There was no apparent effect of intervention on the primary outcome measure of change in global cerebral amyloid (p>0.9). Aerobic exercise was not associated with change in executive function, verbal memory, or visuospatial function (p> = 0.3). Aerobic exercise was not associated with a change in whole brain or hippocampal volume (p>0.1). These results were unchanged when we excluded the subthreshold amyloid group and assessed only those with elevated amyloid (n = 79; S1 Table).

In the per-protocol subset (Table 3; n = 92), there remained a strong physiologic effect of aerobic exercise, with the aerobic exercise group increasing $VO_2$ peak by 12.8%. Again, there was no apparent effect of intervention on primary outcome measure of change in global amyloid burden (p>0.7). Aerobic exercise was not associated with change in executive function, verbal memory or visuospatial function (p>0.2). Aerobic exercise was not associated with a change in whole brain or hippocampal volume (p>0.1). When we examined only those participants with elevated amyloid (n = 65) there were no differences in the results (S1 Table).

**Table 2. Primary outcome measures in the Intent-to-Treat group.**

| Outcome measures | Timepoint | Education (n = 39^) | Aerobic Exercise (n = 78^) | p-value* |
|---|---|---|---|---|
| Global Amyloid Burden (SUVR) | Baseline | 1.2 (0.2) | 1.22 (0.2) | 0.93 |
| | Week 52 | 1.21 (0.2) | 1.22 (0.2) | |
| | Change | 0.01 (0.04) | 0.01 (0.06) | |
| VO$_2$ peak (mL·kg$^{-1}$·min$^{-1}$) | Baseline | 22.7 (5.3) | 21.9 (5.2) | 0.01 |
| | Week 52 | 23.0 (4.9) | 24.3 (5.8) | |
| | Change | 0.1 (2.5) | 2.0 (2.5) | |
| Whole Brain Volume (mL) | Baseline | 1061.7 (114.4) | 1068.7 (109.7) | 0.12 |
| | Week 52 | 1059.1 (115.1) | 1063.4 (109.1) | |
| | Change | -2.6 (-7.2) | -5.3 (-8.7) | |
| Hippocampal Volume (mL) | Baseline | 7.6 (1.0) | 7.5 (0.8) | 0.42 |
| | Week 52 | 7.6 (1.0) | 7.4 (0.8) | |
| | Change | -0.09 (0.14) | -0.07 (0.10) | |
| Executive Function Composite | Baseline | -0.042 (0.365) | 0.029 (0.458) | 0.83 |
| | Week 26 | -0.035 (0.389) | 0.017 (0.625) | |
| | Week 52 | -0.037 (0.452) | 0.018 (0.615) | |
| Verbal Memory Composite | Baseline | 0.032 (0.822) | -0.016 (0.882) | 0.69 |
| | Week 26 | -0.007 (1.014) | 0.003 (0.989) | |
| | Week 52 | 0.051 (0.939) | -0.025 (0.935) | |
| Visuospatial Composite | Baseline | -0.062 (0.572) | 0.031 (0.646) | 0.30 |
| | Week 26 | 0.012 (0.659) | -0.006 (0.715) | |
| | Week 52 | 0.003 (0.559) | -0.001 (0.643) | |

Mean (standard deviation). Cognitive composites at Week 26 and Week 52 can be interpreted as change from baseline.

*2 sample paired t-test comparing baseline and week 52 for amyloid, fitness and volume measures. For cognitive measures, a p-value for treatment by time interaction test from linear mixed models is given.

^Sample size for change in amyloid is Educ:35/Exercise:74. Sample sizes for change in fitness and volumes are Educ:34/Exercise:70 Sample sizes for cognitive measures at baseline, week 26, and week 52 are Educ:39,37,36/Exercise:78,75,75. SUVR = Standard uptake value ratio; VO$_2$ peak = peak oxygen consumption during the graded exercise test.

There were 122 adverse events. Three incidental cardiac findings were discovered at baseline exercise testing and one fall at home during screening for which participants did not receive clearance to continue participation, leaving 118 adverse events following randomization. The education control group had 118 adverse events: 10 mild, 3 moderate and 5 severe, all unrelated to the intervention. The aerobic exercise group had 31 mild (e.g., joint pain resolving with exercise modification), 2 moderate (e.g. joint pain temporarily halting exercise), and 0 severe event related to the intervention, and 48 mild, 12, moderate, and 7 severe events unrelated to the intervention. Examples of mild severity events included seasonal allergies and joint pain resolving with exercise modification. Examples of moderate severity events included outpatient eye surgery and joint pain altering exercise. Examples of severe events included falls at home and hospitalization for gastrointestinal infection. The Data and Safety Monitoring Committee (DSMC) was comprised of 3 physicians unaffiliated with the authors. Adverse events were submitted for review to the DSMC quarterly or within 48 hours if a serious adverse event. Adverse events are summarized in S2 Table.

## Discussion

This is the one of the first randomized controlled trials to prospectively assess the effect of aerobic exercise on cerebral beta-amyloid accumulation in humans. We found no evidence that

**Table 3. Primary outcome measures in the Per-Protocol group.**

| Outcome measure | Timepoint | Education (n = 39^) | Aerobic Exercise (n = 53^) | p-value* |
|---|---|---|---|---|
| Global Amyloid Burden (SUVR) | Baseline | 1.20 (0.2) | 1.23 (0.2) | 0.73 |
| | Week 52 | 1.21 (0.2) | 1.24 (0.2) | |
| | Change | 0.01 (0.04) | 0.01 (0.06) | |
| VO$_2$ peak (mL·kg$^{-1}$·min$^{-1}$) | Baseline | 22.7 (5.3) | 22.7 (5.4) | <0.01 |
| | Week 52 | 23.0 (4.9) | 25.2 (5.8) | |
| | Change | 0.1 (2.5) | 2.5 (2.4) | |
| Whole Brain Volume (mL) | Baseline | 1061.7 (114.4) | 1073.6 (109.3) | 0.12 |
| | Week 52 | 1059.1 (115.1) | 1068.0 (108.3) | |
| | Change | -2.6 (7.2) | -5.5 (7.6) | |
| Hippocampal Volume (mL) | Baseline | 7.6 (1.0) | 7.6 (0.7) | 0.31 |
| | Week 52 | 7.6 (1.0) | 7.5 (0.7) | |
| | Change | -0.09 (0.14) | -0.06 (0.10) | |
| Executive Function Composite | Baseline | -0.042 (0.365) | 0.034 (0.392) | 0.90 |
| | Week 26 | -0.035 (0.389) | 0.041 (0.606) | |
| | Week 52 | -0.037 (0.452) | 0.012 (0.607) | |
| Verbal Memory Composite | Baseline | 0.032 (0.822) | 0.024 (0.861) | 0.47 |
| | Week 26 | -0.007 (1.014) | -0.039 (0.931) | |
| | Week 52 | 0.051 (0.939) | -0.053 (0.839) | |
| Visuospatial Composite | Baseline | -0.062 (0.572) | 0.067 (0.657) | 0.22 |
| | Week 26 | 0.012 (0.659) | 0.036 (0.727) | |
| | Week 52 | 0.003 (0.559) | 0.017 (0.669) | |

Mean (standard deviation). Cognitive composites at Week 26 and Week 52 can be interpreted as change from baseline.

* For amyloid, fitness and brain volume measures, a p-value from ordinary least squares regression adjusted for sex, age, education, and amyloid status comparing the change (baseline to week 52) between the two groups is given. For cognitive measures, a p-value for treatment by time interaction test from linear mixed models adjusted for sex, age, education, and amyloid status among per protocol subgroup is given.

^Sample size for amyloid at baseline and week 52 are Educ:39,35/Exercise:53,53. Sample size for VO$_2$ at baseline and week 52 are Educ:39,34/Exercise:53,53. Sample size for brain volumes at baseline, week 52 are Educ:34,34/Exercise:52,52. Sample sizes for cognitive measures at baseline, week 26, and week 52 are Educ:39,37,36 /Exercise:53,53,53. SUVR = Standard uptake value ratio; VO$_2$ peak = peak oxygen consumption during the graded exercise test.

one year of aerobic exercise influences cerebral amyloid burden in a cohort of cognitively normal participants with elevated and subthreshold levels of amyloid, individuals who are at highest risk of clinically significant amyloid accumulation. We did find significant and meaningful changes in cardiorespiratory fitness suggesting the intervention was of sufficient intensity and duration to provoke physiologic effects. Despite this, however, we did not find aerobic exercise effects on whole brain volume, hippocampal volume, or cognitive measures. Our observed atrophy rates were consistent with those previously reported in cognitively normal older adults [60]. We believe these null findings support a hypothesis that the widely reported brain benefits of exercise are modest and driven mechanistically by the mitigation of non-amyloid pathologies.

It is important to consider the context of our findings in a highly selected sample that likely skewed towards fewer age-related pathologies, such as subclinical cerebrovascular disease, than most studies in the literature. We assessed over 1,500 participants for eligibility (see Fig 1) and excluded those with cardiopulmonary concerns and systemic illnesses while retaining those (largely through participant self-selection) interested in potentially participating in a year of rigorous exercise. Importantly, we performed careful clinical and cognitive assessments to exclude those with cognitive impairment, despite the presence of cerebral amyloid; thus this group is likely enriched with unmeasured (and currently poorly defined) resilience factors,

such as the absence of cerebrovascular disease or other age related pathologies. The lack of observed exercise effects on amyloid, cognitive, or brain structure outcomes despite clear exercise related effects on physiologic outcomes (cardiorespiratory fitness) leads us to hypothesize that the brain benefits of aerobic exercise observed widely in the literature are not driven by effects on AD pathology but instead are likely driven by the mitigation of aging related vascular or other non-amyloid pathologies. Indeed, recent work has identified cerebrovascular outcomes and important mediators of cognitive change following exercise [5,14,61].

It remains possible that our results are related to Type II error where a true effect is obscured by lack of power or methodological issues. Our null finding for an effect of aerobic exercise on amyloid accumulation is surprising given the number of animal exercise studies reporting reduced amyloid accumulation rates and lower amyloid loads [16,17,19–21]. However, small human intervention studies have examined the impact of exercise on amyloid with inconclusive results. At least three intervention studies have examined the impact of exercise on serum amyloid concentration, with none reporting reliable reductions in amyloid as a consequence of exercise [62–64]. The amyloid tracer we employed (18F-AV45) may lack sufficient sensitivity/specificity to index subtle changes in amyloid induced by exercise in cognitively normal older adults. In the overall group (n = 106) we observed a 0.8% (SD 4.4%) increase in amyloid compared to reported annual changes of 1–4%, a range influenced by where an individual is on the sigmoid curve of accumulation over the lifespan [65]. Additionally, when examining our subgroups, the elevated group had a 1.5% (SD 4.5%) annual rate of accumulation compared to a decline of -0.9% (SD 3.6%) in subthreshold group, a decline that was not in line with our expectations for this group. Recent serial amyloid PET studies suggest that reference region selection (i.e., whole cerebellum vs cerebellar white matter) can influence measured change over time and that annual participant scan variance may be higher than the expected annual rate of amyloid PET change, especially when only two data points are present [66]. However, the tracer can sufficiently track dose-related amyloid change in investigational medication trials [67], suggesting that if our failure to observe changes was related to measurement error, exercise is unlikely to have a large effect on cerebral amyloid levels.

It is possible that our inclusion of individuals in the subthreshold range (n = 38) who were not elevated reduced our ability to detect reductions in amyloid by enhancing a floor effect. However, when assessing only those in the elevated group (n = 79), there were no trends suggesting an effect of aerobic exercise on amyloid accumulation. Additionally, the potential benefits of aerobic exercise to influence cerebral amyloid may require a longer duration than 52 weeks. One year may not be long enough to meaningfully alter amyloid levels or the rate of accumulation. Future studies looking at more than two amyloid PET time points to reduce scan to scan variance and longer time interval (at least 2 years) may be important to investigate whether exercise can impact the rate of amyloid accumulation. The non-significantly higher proportion of E4 carriers in the treatment group may have subtly impacted cognitive decline and amyloid accumulation, potentially obscuring our ability to detect a benefit of the intervention. As a sensitivity analysis, we also tested our models with SUVR as a covariate, and with APOE4 and APOE4 by Treatment Arm as factors. There was no appreciable change in our results in these analyses (data not shown). Though we detected no difference in carrier versus non-carrier performance, brain volume change, or amyloid accumulation, over 1-year, future studies may wish to consider E4 carriage as a blocking variable for randomization

Our lack of effect on our secondary outcomes of brain volume and cognitive performance was surprising, especially given the strong physiologic effects of the exercise intervention on cardiorespiratory fitness. Practice effects, especially in cognitively normal older adults, reduce power to discern group differences in cognitive performance [68,69] but despite this, a number of well-designed RCTs have shown that aerobic exercise benefits cognition in older adults,

though not specifically in those with elevated cerebral amyloid [2–5,50]. Many studies have also demonstrated benefits to whole brain gray matter and hippocampal volume, with one notable study reporting a decrease in whole brain gray matter volume after 12 months of resistance training [70]. We have previously suggested that cardiorespiratory fitness gains are critical for cognitive or brain improvements [4,49,71]. Simply exercising without increasing cardiorespiratory fitness, and therefore eliciting associated physiological and biochemical adaptations, does not appear to support brain or cognitive changes. Despite significantly increasing maximal cardiorespiratory capacity in this trial, we did not identify the same relationship in the present study. This may suggest that those with elevated amyloid are more resistant to the putative brain benefits of aerobic exercise.

There are several additional limitations to this study. Our sample was almost exclusively White, non-Hispanic and highly educated. This severely limits the generalizability of our findings and highlights structural racism and inequity related to clinical trial access. As a result, we have begun assessing the design of our trials and increased our efforts to inclusively design our exercise trials with and for underrepresented communities [72–75]. An additional limitation is the use of self-reported physical activity versus an activity monitor and lack of a treatment fidelity analysis. It is possible that exercise activity increased in the control group. However, consistent with participant self-report we saw evidence of fitness change only in the exercise group. Finally, it may be possible that the selected dose of exercise (duration and intensity) is insufficient or ill-suited to change amyloid accumulation. Future work should consider resistance training, or alternate intensities.

It is critical to note that the results of the study do not suggest that aerobic exercise is not beneficial. Aerobic exercise continues to have tremendous and unquestionable benefits for the body. Potential mechanisms for benefits observed with exercise include the upregulation of proteins involved in the clearance of amyloid [25,76] and reduction of systemic inflammation [77]. Tailoring exercise prescription may maximize the engagement of these processes. Our aerobic exercise group, for example, increased $VO_2$ peak by 11%, with 11 individuals moving from a state of potentially impaired independence with a $VO_2$ peak below 20 mL·kg$^{-1}$·min$^{-1}$ [78], to a more fully functional cardiorespiratory state of a $VO_2$ peak above 20 mL·kg$^{-1}$·min$^{-1}$. Only one individual in the control group made that positive change, whereas 5 individuals in the group dropped below a $VO_2$ peak of 20 mL·kg$^{-1}$·min$^{-1}$ during the study.

## Conclusions

The results of this trial do not support the hypothesis that 52 weeks of aerobic exercise influences amyloid burden in cognitively normal older adults. Additionally, secondary outcomes did not support prior work indicating that aerobic exercise benefits measures of brain health or cognition, at least in a cohort of cognitively normal older adults at elevated risk for Alzheimer's due to elevated cerebral amyloid burden. The observed lack of cognitive or brain structure benefits, despite strong systemic cardiorespiratory effects of the intervention, suggests brain benefits of exercise reported in other studies are likely to be related to non-amyloid effects.

A large-scale, definitive trial is currently underway which will help to confirm or refute these findings [6].

## Supporting information

**S1 Checklist. CONSORT 2010 checklist of information to include when reporting a randomised trial**\*.
(DOC)

**S1 Table. Primary and secondary outcomes of individuals with elevated amyloid.** Mean and standard deviation. Mean and standard deviation. ^ Sample size for amyloid change Educ:25/Exercise:49. Sample size for change in VO$_2$ peak Educ:24/Exercise:49. Sample size for change in brain volumes Educ:24/Education:47. Sample size for change in cognitive measures at baseline, week 26 and week 52 are Educ:26,26,25/Education:52,51,50. SUVR = Standard Uptake Value Ratio; VO$_2$ peak = peak oxygen consumption during graded exercise test. * For amyloid, fitness and brain volume measures, a p-value from ordinary least squares regression adjusted for sex, age, and education comparing the change (baseline to week 52) between the two groups is given. For cognitive measures, a p-value for treatment by time interaction test from linear mixed models adjusted for sex, age, education, and amyloid status among per protocol subgroup is given.
(DOCX)

**S2 Table. Adverse events.**
(DOCX)

**S1 File.**
(DOCX)

## Acknowledgments

We wish to thank the participants who gave their time for this study. We also wish to thank the YMCA of Greater Kansas City and Genesis Health Clubs for supporting community-based exercise research.

## Author Contributions

**Conceptualization:** Eric D. Vidoni, Jill K. Morris, Jonathan Mahnken, Robyn Honea, David K. Johnson, Jeffrey M. Burns.

**Data curation:** Eric D. Vidoni, Jill K. Morris, Amber Watts, Jonathan Mahnken, Suzanne L. Hunt.

**Formal analysis:** Eric D. Vidoni, Jill K. Morris, Amber Watts, Mark Perry, Ashwini S. Kamat, Jonathan Mahnken, Ryan Townley, Ashley R. Shaw, James Vacek, Jeffrey M. Burns.

**Funding acquisition:** Jonathan Mahnken, Jeffrey M. Burns.

**Investigation:** Eric D. Vidoni, Jill K. Morris, Amber Watts, Jon Clutton, Angela Van Sciver, Ashwini S. Kamat, Jonathan Mahnken, Suzanne L. Hunt, Robyn Honea, Ashley R. Shaw, David K. Johnson, James Vacek, Jeffrey M. Burns.

**Methodology:** Eric D. Vidoni, Jill K. Morris, Mark Perry, Jon Clutton, Suzanne L. Hunt, Robyn Honea, Jeffrey M. Burns.

**Project administration:** Eric D. Vidoni, Jill K. Morris, Jon Clutton, Angela Van Sciver, Robyn Honea, David K. Johnson, James Vacek, Jeffrey M. Burns.

**Resources:** Jeffrey M. Burns.

**Software:** Eric D. Vidoni, Robyn Honea.

**Supervision:** Eric D. Vidoni, Jon Clutton, Angela Van Sciver, Jeffrey M. Burns.

**Writing – original draft:** Eric D. Vidoni, Jill K. Morris, Amber Watts, Mark Perry, Jon Clutton, Jonathan Mahnken, Suzanne L. Hunt, Ryan Townley, Robyn Honea, Ashley R. Shaw, Jeffrey M. Burns.

**Writing – review & editing:** Eric D. Vidoni, Jill K. Morris, Amber Watts, Mark Perry, Jon Clutton, Angela Van Sciver, Ashwini S. Kamat, Jonathan Mahnken, Suzanne L. Hunt, Ryan Townley, Robyn Honea, Ashley R. Shaw, David K. Johnson, James Vacek, Jeffrey M. Burns.

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
