## [Decision Letter · Decision Letter 0]

9 Oct 2020

PONE-D-20-24208

Effect of Aerobic Exercise on Amyloid Accumulation in Preclinical Alzheimer’s: A 1-Year Randomized Controlled Trial

PLOS ONE

Dear Dr. Burns,

Thank you for submitting your manuscript to PLOS ONE. After careful consideration, we feel that it has merit but does not fully meet PLOS ONE’s publication criteria as it currently stands. Therefore, we invite you to submit a revised version of the manuscript that addresses the points raised during the review process.

We look forward to receiving your revised manuscript.

Kind regards,

Ashley I Bush, MD PhD

Academic Editor

PLOS ONE

Journal Requirements:

'Federalwide Assurance (FWA) has been approved for KUMC. The FWA covers two IRBs which operate at the KUMC main campus in Kansas City as well as other IRBs designated by the KUMC HRPP as reviewing IRBs.

Institutional Organization #: IORG0000100

Federalwide Assurance #: FWA00003411

KUMC IRBs

IRB 1#: IRB00000161

IRB 3#: IRB00006196

University of Kansas Medical Center Human Subjects committee (HSC#13376) written informed consent was obtained from all participants.'

a. Please amend your current ethics statement to confirm that your named institutional review board or ethics committee specifically approved this study.

'This work was supported by the National Institutes of Health R01 AG043962 (JMB); K99 AG050490 (JKM) and gifts from Frank and Evangeline Thompson (JMB), The Ann and Gary Dickinson Family Charitable Foundation, John and Marny Sherman, and Brad and Libby Bergman. Institutional infrastructure support for testing was provided in part by UL1 TR000001 (RJB) and P30 AG035982 (RHS JMB). Lilly Pharmaceuticals provided a grant to support F18-AV45 doses and partial scan costs (JMB). The content is solely the responsibility of the authors and does not necessarily represent the official views of the National Institutes of Health. The funders had no role in study design, data collection and analysis, decision to publish, or preparation of the manuscript.'

We note that you received funding from a commercial source: Lilly Pharmaceuticals

Reviewers' comments:

Reviewer's Responses to Questions

**Comments to the Author**

1. Is the manuscript technically sound, and do the data support the conclusions?

Reviewer #1: Yes

Reviewer #2: Partly

Reviewer #3: Yes

Reviewer #4: Yes

2. Has the statistical analysis been performed appropriately and rigorously? 

Reviewer #1: Yes

Reviewer #2: No

Reviewer #3: Yes

Reviewer #4: Yes

3. Have the authors made all data underlying the findings in their manuscript fully available?

Reviewer #1: Yes

Reviewer #2: No

Reviewer #3: Yes

Reviewer #4: Yes

4. Is the manuscript presented in an intelligible fashion and written in standard English?

Reviewer #1: Yes

Reviewer #2: Yes

Reviewer #3: Yes

Reviewer #4: Yes

5. Review Comments to the Author

Reviewer #1: This is a well-written manuscript with appropriate methodology. The efficacy of lifestyle interventions to reduce dementia risk is of profound import to society given the burden of neurocognitive disorders world-wide.

Inclusion of amyloid PET is a particular strength of this study. Unfortunately, the MRI protocol was rather limited, and it would have been nice if vascular burden had been measured, e.g. with T2/FLAIR sequences. This would have facilitated an analysis of whether exercise had any effect on cerebrovascular burden, which is of interest given the lack of impact on amyloid accumulation in this study as measured by florbetapir PET.

In line 276, the authors report that there were no differences across intervention groups, as shown in Table 1. Would the authors please clarify if they mean no significant differences, and at what p value? Of particular note here is the higher proportion of ApoE E4 carriers in the aerobic exercise group. In theory, a significantly higher proportion of ApoE E4 alleles in the treatment group could increase the risk of cognitive decline and amyloid accumulation in these subjects, thus potentially reducing the likelihood of detecting a true benefit of the intervention.

The fact that an effect of exercise on measures of amyloid or cognitive performance was not found does not diminish the importance of this study. The authors provide a comprehensive discussion about potential weaknesses. The risk of a Type II error is especially important given the sample size and duration of intervention. The authors also appropriately comment that brain benefits of exercise (which were not demonstrated in this study) may well be mediated by mechanisms other than an effect on amyloid. This is particularly important given the large number of amyloid-lowering clinical trials to date that have failed to meet primary endpoints. The argument for considering aspects of neuropathology aside from amyloid in Alzheimer's disease is getting stronger. The authors address the relevance of their findings to this body of work in a sensible fashion, concluding that their null findings support a hypothesis that the reported brain benefits of exercise are modest and may be driven by factors other than the mitigation of amyloid pathology.

In summary, this is an appropriately-designed, well-written study with important findings. I recommend that this manuscript is published, once the authors have clarified the issue mentioned above regarding significance of difference in line 276 and Table 1.

Reviewer #2: The authors performed a RCT to investigate the effect of a 52-week supervised exercise intervention on markers of brain health including brain volume (whole brain and hippocampal), cognitive assessments and cerebral amyloid (as measured with 18F-AV45 PET) in a group of 117 cognitively normal older, almost exclusively white and educated (only 1 African American in the intervention group), adults with elevated (n=79) and sub-threshold (n=38) levels of cerebral amyloid. They compared the effects of 150 minutes per week of aerobic exercise to an education control intervention. Participants were tested at baseline, 26 weeks, and 52 weeks. Cardiorespiratory fitness testing was performed at baseline and Week 52 to assess response to exercise.

One hundred and ten participants completed the study. The aerobic exercise group significantly improved cardiorespiratory fitness (11% vs. 1% in the control group) but there were no differences in change measures of amyloid, brain volume, or cognitive performance compared to control.

They found that aerobic exercise was not associated with reduced amyloid accumulation in cognitively normal older adults with cerebral amyloid. They conclude that the brain benefits of exercise are likely to be related to non-amyloid effects.

.

I have some other questions and comments for the authors:

1. Note that the data access statement is incomplete in this submission.

2. Introduction: “Higher levels of aerobic fitness are associated with age-related change in brain volume and cognition at both cross-section and over time.” This is true but is very diffuse. Given that this is the hypothesis for this study I think specific detail is required here.

3. Was sedentary/under active an inclusion or exclusion criterion? This needs to be stated in the abstract.

4. How could this be a convenience sample? This is a RCT?

5. What did you compare your study group to in terms of annual amyloid accumulation? Did you already know the expected increase in amyloid for your participants? If so, please state.

6. Please give more information on how you defined a study certified exercise facility.

7. What training did you give to the personal trainers? How did you ensure blinding? Did different EPs perform the fitness testing?

8. Did you perform a treatment fidelity analysis? If so, when and with how many participants?

9. Minor point but why not an isotropic MPRAGE acquisition?

10. Line 242 there is a double full-stop.

11. Your intervention was predicated on an effect size of 40%. That’s large. What evidence did you have for this? Most exercise interventions have small-medium effect sizes, depending on the outcome. It is highly likely that your study is very underpowered. You state “93% power to detect this conservative 265 anticipated effect of exercise on amyloid burden”. This is not conservative at all.

12. What was the rationale for increasing the sample to 120? Surely the expected effect size on people with sub-threshold amyloid would be substantially less.

13. Why use self-report of exercise? It would have been easy to fit your subjects with a physical activity monitor. It is possible that exercise increased in both and that this affected their amyloid.

14. Apart from fitness, did you measure any other cardiovascular effects? BP, change in BMI, blood cholesterol, etc.?

15. In Table 2, you state: “^Sample size for change in amyloid is Educ:35/Exercise:74. Sample sizes for change in fitness and volumes are Educ:34/Exercise:70 Sample sizes for cognitive measures at baseline, week 26, and week 52 are educ:39,37,36/Exercise:78,75,75.” Don’t you mean number of participants?

16. 122 is a lot of AEs related to aerobic exercise. Please include a table of these. Also, include details of the safety monitoring committee, how these were reported and adjudicated, how many SAEs, etc.

17. Discussion, line 337-8: “Despite this, however, individuals with elevated levels of amyloid appeared resistant to aerobic exercise effects on whole brain volume, hippocampal volume, or cognitive measures.” You can’t state that. You can only state that you found no difference.

18. Line 348: “likely enriched with unmeasured (and currently poorly defined) resilience factors”. Surely the fact that they were all white and educated should be discussed?

19. The discussion should be abbreviated given the fact you were likely very underpowered. You may need hundreds in each arm of the study to show difference in your chosen outcomes at 12 months. It would be important to publish the natural history of change in amyloid as measured by your tracer.

Reviewer #3: Vidoni et al. have produced a well-written manuscript reporting findings from a 12-month RCT investigating the effect of aerobic exercise on amyloid accumulation, MRI, fitness, and cognitive outcomes in pre-clinical AD.

Beyond the impact of the intervention on fitness, the manuscript reports null findings, which are in my opinion, as important a contribution to the literature as positive results.

The following comments and suggestions are listed by manuscript section:

Introduction:

Line 65 "Higher levels of aerobic fitness are associated with age-related change in brain volume and cognition at both cross-section and over time.[3, 9-13]" - please be explicit regarding the nature of the relationship.

Line 67 "In randomized controlled trials, aerobic exercise promotes brain plasticity and attenuates hippocampal atrophy while improving spatial memory.[3, 5, 14, 15]" - ref 15 reports increased hippocampal volume rather than attenuated atrophy; please amend sentence accordingly.

Line 79 "...cognitively healthy adults..." - the term cognitively normal is used everywhere else in the manuscript, suggest amending for consistency.

"...and those at high genetic risk for AD.[26, 33, 34]" - ref 29 needs adding here as the reported benefits of exercise on brain amyloid are in APOE e4 carriers.

Line 91-94 "Individuals with subthreshold levels of cerebral amyloid (individuals with non-elevated amyloid

PET but with quantitative measures near the threshold for being elevated) may be more likely to

accumulate amyloid and have memory decline,[36] suggesting they are good candidates for

prevention studies.[37]" - to be "subthreshold" these individuals must have already accumulated some amyloid so the wording of "may be more likely to accumulate amyloid" needs amending.

Line 98 "..those with subthreshold levels of cerebral amyloid" - needs amending to '...those who are CN but with subthreshold levels of cerebral amyloid' (CN is implicit in preclinical AD in the earlier part of the sentence, but not in the subthreshold group).

Line 98-99 "We hypothesized that 52 weeks of aerobic exercise would be associated with reduced amyloid burden, reduced hippocampal atrophy,..." - "reduced amyloid burden", or a slowing of amyloid accumulation?

Methods:

Line 176 "Heart Rate Reserve" - change to HRR as defined on the previous line.

Line 247-248 "...other covariates (age, sex, education, and PET amyloid status [elevated vs. subthreshold])..." - I am extremely interested to know if the authors controlled for baseline continuous SUVR rather than the categorical of elevated/subthreshold in any analyses undertaken that have not been reported here. As the authors rightly point out in the Discussion, amyloid accumulation is a sigmoidal curve and I do wonder if combining elevated and subthreshold may reduce the sensitivity of the analysis (particularly when combined with the test/re-test variability of FBP, and having only two amyloid scans).

Results:

Line 280 Table 1 - please change ApoE to APOE to reflect gene rather than protein.

Also, this is the first and only time that APOE is mentioned in the entire manuscript; consequently it needs defining and explaining.

On this topic, I am curious as to why APOE e4 status hasn't been included anywhere in the analysis, particularly given this statement on Line 77 "...greater amounts of self-reported physical activity (i.e., volitional

behavior that is part of daily function) is associated with evidence of lower cerebral amyloid levels among cognitively healthy adults,[26-32] and those at high genetic risk for AD." where the published studies on "high genetic risk for AD" include APOE e4. Plus, we know that APOE e4 impacts amyloid accumulation, and cognitive performance (particularly in preclinical AD). Therefore, I believe this additional analysis is certainly warranted.

Table 1 - no need to include % in columns when % is included in the row header. Please also define abbreviations in the Table footnotes.

Table 2 - Is "Intent-to-Treat" needed twice in the Table heading?

Global amyloid burden / change / 0.01 (.04), missing a '0'. Please also define abbreviations in the Table footnotes.

Table 3 - Is "Per-Protocol" needed twice in the Table heading?

Curious that the loss of brain volume in the Exercise Group is twice that of the Control Group - I wonder if this is related to, the higher % of APOE e4 carriers in the Exercise Group, where individuals are on the sigmoidal amyloid accumulation curve, or both (another reason why it could be insightful to include APOE e4 and continuous SUVR in the analysis). Please also define abbreviations in the Table footnotes.

Discussion:

Line 333 "... individuals who are at highest risk of amyloid accumulation." - see earlier comment regarding Line 91-94.

Line 336-340 "Despite this, however, individuals with elevated levels of amyloid appeared resistant to aerobic exercise effects on whole brain volume, hippocampal volume, or cognitive measures. We believe these null findings support a hypothesis that the widely reported brain benefits of exercise are modest and driven mechanistically by the mitigation of non-amyloid pathologies." - this is most certainly a possibility. However, some consideration should also be given in the Discussion to the fact that it is also possible that the selected aerobic intervention wasn't effective (despite the improvements in fitness observed). Perhaps higher intensity of aerobic exercise is needed to slow amyloid accumulation?

Line 342 "...skewed towards fewer age-related pathologies, such as subclinical cerebrovascular disease,..." - perhaps; although it is also highly likely that your sample includes individuals with vascular amyloid deposits.

Line 362 sudden introduction of the term "Aβ" whilst "amyloid" has been used thus far - either need to define or stick with amyloid.

Line 381 "Additionally, the potential benefits of aerobic exercise to influence cerebral amyloid may require a longer duration than 52 weeks." - or more intense aerobic exercise?

General Comments:

Please pay close attention to the comments above around factoring in APOE e4 and baseline continuous SUVR into the analysis.

I also wonder if the authors have considered whether there are indirect effects of the intervention on the outcome measures that relate to improvements in cardiorespiratory fitness? Specifically, there were no observed effects on amyloid, MRI and cognitive outcomes when comparing group performance from pre- to post-intervention. However, it is possible that changes in cardiorespiratory fitness from pre- to post-intervention are associated with changes in the outcome measures. It is also possible that APOE genotype may moderate these effects.

For a later date, it may well also be valuable to assess the impact of the intervention of blood levels of amyloid species using the new generation of ultra sensitive assays that appear to be extremely promising.

Reviewer #4: This is an excellent study, clearly reported. I have a few minor comments.

The authors should give details of how the randomisation schedule was produced, eg. whether blocking or stratification were used etc.

I think that the first sentence of the statistical analysis section should read ‘…including means, standard deviations and…’

Statistical analysis, second sentence: two-sample t-tests and paired t-tests are different things. What the authors should have used here are two-sample t-tests comparing the change from baseline between the two groups. I am not sure which they have actually used.

The intention to treat and per protocol populations should be defined in the methods rather than in the results.

Figure 1 is of poor resolution.

The results from Tables 2 and 3 are reported in the text only in terms of their p-values. It would be helpful for the authors to discuss the clinical significance of the differences, even where there is a lack of statistical significance. Eg. Whole brain volume has a p-value >0.05 but there does seem to have been more of a decrease in the exercise group. Is the magnitude of the difference clinically significant? (I am a statistician and I genuinely do not know if the drops in whole brain volume and the magnitude of the difference between the groups is clinically relevant).

Table 2: number formats should be the same within outcomes, eg. Baseline global amyloid should be 1.20 rather than 1.2 and the SE of the change should be 0.04, not .04.

6. PLOS authors have the option to publish the peer review history of their article (what does this mean?). If published, this will include your full peer review and any attached files.

Reviewer #1: No

Reviewer #2: No

Reviewer #3: No

Reviewer #4: **Yes: **Sarah J.E. Barry

---

## [Author Response · Author response to Decision Letter 0]

5 Nov 2020

Dr. Bush,

Academic Editor, PLOS One

We appreciate the thoughtful reviews of our manuscript PONE-D-20-24208, “Effect of Aerobic Exercise on Amyloid Accumulation in Preclinical Alzheimer’s: A 1-Year Randomized Controlled Trial”. We have carefully considered each critique and responded. Critiques are listed and numbered, and the responses begins with “>>”. In some instances, we have provided text changes in the response document as well for the ease of the Reviewer. These are in “quotes”. 

We have provided a markup copy as well as a clean copy of our revised manuscript. We thank the reviewers for helping us to improve the manuscript and look forward to positive reception.

Sincerely,

Jeffrey M. Burns, MD, MS

University of Kansas Medical Center.

Editorial Review:

>>We have adjusted our formatting and file names to our best interpretation of the instructions.

2.Please amend your current ethics statement to confirm that your named institutional review board or ethics committee specifically approved this study. Once you have amended this/these statement(s) in the Methods section of the manuscript, please add the same text to the “Ethics Statement” field of the submission form (via “Edit Submission”).

>>We have amended our Ethics Statement in the submission and manuscript to read: 

“University of Kansas Medical Center Human Subjects committee approved the protocol (HSC#13376) and written informed consent was obtained from all participants.”

>>We have included our competing interests statement in the cover letter with the additional statement that "This does not alter our adherence to PLOS ONE policies on sharing data and materials.” 

4. PLOS requires an ORCID iD for the corresponding author in Editorial Manager on papers submitted after December 6th, 2016. Please ensure that you have an ORCID iD and that it is validated in Editorial Manager. 

>>We have linked the ORCID iD of the corresponding author in Editorial Manager.

Reviewer #1: 

1. In line 276, the authors report that there were no differences across intervention groups, as shown in Table 1. Would the authors please clarify if they mean no significant differences, and at what p value? Of particular note here is the higher proportion of ApoE E4 carriers in the aerobic exercise group. In theory, a significantly higher proportion of ApoE E4 alleles in the treatment group could increase the risk of cognitive decline and amyloid accumulation in these subjects, thus potentially reducing the likelihood of detecting a true benefit of the intervention.

>>We have amended the line per the Reviewer’s recommendation and added a “p-value” column to Table 1. We agree it’s possible that the non-significant difference in E4 carriage between the two groups could adversely impact the interpretation of our outcomes. However, in response to other reviewers we also conducted some secondary sensitivity analyses and found no interaction of treatment group and E4 carriage on our outcomes. Nevertheless, we have added this as a potential future direction.

“The non-significantly higher proportion of E4 carriers in the treatment group may have subtly impacted cognitive decline and amyloid accumulation, potentially obscuring our ability to detect a benefit of the intervention. Though we detected no difference in carrier versus non-carrier performance, brain volume change, or amyloid accumulation, over 1-year, future studies may wish to consider E4 carriage as a blocking variable for randomization.”

2. The fact that an effect of exercise on measures of amyloid or cognitive performance was not found does not diminish the importance of this study. The authors provide a comprehensive discussion about potential weaknesses. The risk of a Type II error is especially important given the sample size and duration of intervention. The authors also appropriately comment that brain benefits of exercise (which were not demonstrated in this study) may well be mediated by mechanisms other than an effect on amyloid. This is particularly important given the large number of amyloid-lowering clinical trials to date that have failed to meet primary endpoints. The argument for considering aspects of neuropathology aside from amyloid in Alzheimer's disease is getting stronger. The authors address the relevance of their findings to this body of work in a sensible fashion, concluding that their null findings support a hypothesis that the reported brain benefits of exercise are modest and may be driven by factors other than the mitigation of amyloid pathology.

>>We appreciate this thoughtful and well-stated perspective. 

Reviewer #2: 

1. Note that the data access statement is incomplete in this submission.

>>We have revised our data access statement and have archived the data for public availability. 

“All files are available from the Harvard Dataverse database (Vidoni, Eric, 2020, "Alzheimer's Prevention Through Exercise (APEx) - NCT02000583", https://doi.org/10.7910/DVN/B9I1F8, Harvard Dataverse, V1, UNF:6:pgNe0pp64djpi7AAdwhBiw== [fileUNF]).”

2. Introduction: “Higher levels of aerobic fitness are associated with age-related change in brain volume and cognition at both cross-section and over time.” This is true but is very diffuse. Given that this is the hypothesis for this study I think specific detail is required here.

>>Thank you for this comment. We have amended the sentence to read:

“Higher levels of aerobic fitness are associated with age-related improvements or attenuated decline in brain volume and cognition at both cross-section and over time”

3. Was sedentary/under active an inclusion or exclusion criterion? This needs to be stated in the abstract.

>>We have added this criterion to the abstract.

4. How could this be a convenience sample? This is a RCT?

>>Our participants were drawn from individuals who self-selected to present for exercise research. We did not purposively sample to recruit and match regional demographics. As such, the participant population was that which was “close at hand”. The participants were indeed randomized, but we cannot claim that they accurately represent the larger population within those randomly assigned groups.

5. What did you compare your study group to in terms of annual amyloid accumulation? Did you already know the expected increase in amyloid for your participants? If so, please state.

>>Amyloid accumulation was calculated in units normalized to cerebellar amyloid signal (SUVR). Change over the 52-week study was calculated as the individual difference from baseline to week 52 global amyloid signal (noted in our Outcomes section) We did not attempt to compare amyloid signal between our participants and previously published 18F-AV45 accumulation rates.

6. Please give more information on how you defined a study certified exercise facility.

>>Study certified exercise facilities are those which have undergone extensive training and vetting for appropriate safety and privacy controls. As noted in the manuscript we have published our method for delivering community-based exercise protocols with fidelity and adherence (Vidoni et al. Contemp Clin Trials 2012).

7. What training did you give to the personal trainers? How did you ensure blinding? Did different EPs perform the fitness testing?

>>Personal trainers were not blinded to intervention as they were required to oversee exercise. As no doubt the Reviewer knows, EPs and personal trainers are not the same profession. Personal Trainers provided the oversight and guidance on exercise at community locations. Exercise Physiologists were employed by the institution to oversee exercise testing. Neither the trainers nor the raters had any interaction with participants beyond their respective roles, which did not overlap. This allowed the raters to remain blinded to the intervention. 

We have revised these sections to make it clear that the trainers were employed by the exercise facility and raters had no interaction with participants beyond their appointed testing visits.

8. Did you perform a treatment fidelity analysis? If so, when and with how many participants?

>>We did not. We have added this as a limitation (see point 13) and we appreciate the suggestion for our future studies.

9. Minor point but why not an isotropic MPRAGE acquisition?

>>We have adopted ADNI imaging protocols for nearly all our studies to allow for maximum shareability. At the time of study start-up our protocol matched the ADNI2 protocol to the extent our scanner would allow.

10. Line 242 there is a double full-stop.

>>We have amended this as suggested.

11. Your intervention was predicated on an effect size of 40%. That’s large. What evidence did you have for this? Most exercise interventions have small-medium effect sizes, depending on the outcome. It is highly likely that your study is very underpowered. You state “93% power to detect this conservative 265 anticipated effect of exercise on amyloid burden”. This is not conservative at all.

>>At the time of study inception, amyloid imaging was still very new with little clarity on how 18F-AV45 would respond to intervention. In the Statistical Analysis section we noted multiple effect sizes from pharmaceutical trials, at that time, the only work we could base our study on. We extensively discuss the possibility of Type II error in the Discussion.

12. What was the rationale for increasing the sample to 120? Surely the expected effect size on people with sub-threshold amyloid would be substantially less.

>>It’s not clear that the effect on subthreshold individuals would be less in our view. In fact, there is substantial evidence of an S-curve in amyloid accumulation, which would suggest that subthreshold individuals experience greater accumulation rates, and perhaps the greatest amyloid removal rates if possible. The sample size of 120 was a scientific and financial compromise.

13. Why use self-report of exercise? It would have been easy to fit your subjects with a physical activity monitor. It is possible that exercise increased in both and that this affected their amyloid.

>>In retrospect, the use of a monitor would have been appropriate. It is possible that exercise increased in the control group. However, we saw no such evidence of fitness change consistent with self-report. This validates our belief that only the intervention group significantly increased their exercise activity. However, we have added this as a limitation in the Discussion.

“An additional limitation is the use of self-reported physical activity versus an activity monitor and lack of a treatment fidelity analysis. It is possible that exercise activity increased in the control group. However, consistent with participant self-report we saw evidence of fitness change only in the exercise group.”

14. Apart from fitness, did you measure any other cardiovascular effects? BP, change in BMI, blood cholesterol, etc.?

>>In following CONSORT best practice, we have limited this report to our primary outcome and prespecified secondary outcomes of interest. If we analyze at these important tertiary outcomes in future manuscripts we will clearly state that these are secondary outcomes.

15. In Table 2, you state: “^Sample size for change in amyloid is Educ:35/Exercise:74. Sample sizes for change in fitness and volumes are Educ:34/Exercise:70 Sample sizes for cognitive measures at baseline, week 26, and week 52 are educ:39,37,36/Exercise:78,75,75.” Don’t you mean number of participants?

>>We have made the requested change.

16. 122 is a lot of AEs related to aerobic exercise. Please include a table of these. Also, include details of the safety monitoring committee, how these were reported and adjudicated, how many SAEs, etc.

>>We appreciate the opportunity to revise this section as we were not clear that this indicates the full breadth of adverse events, not just those related to the intervention. We have revised this section and added a supplementary table, S2 Table, for clarity.

17. Discussion, line 337-8: “Despite this, however, individuals with elevated levels of amyloid appeared resistant to aerobic exercise effects on whole brain volume, hippocampal volume, or cognitive measures.” You can’t state that. You can only state that you found no difference.

>>We have amended the sentence as requested.

“Despite this, however, we did not find aerobic exercise effects on whole brain volume, hippocampal volume, or cognitive measures.”

18. Line 348: “likely enriched with unmeasured (and currently poorly defined) resilience factors”. Surely the fact that they were all white and educated should be discussed?

>>Thank you. This is an extremely important point. We have added the following limitation section to our Discussion.

“Our sample was almost exclusively White, non-Hispanic and highly educated. This severely limits the generalizability of our findings and highlights structural racism and inequity related to clinical trial access. As a result we have begun assessing the design of our trials and increased our efforts to inclusively design our exercise trials with and for underrepresented communities (Vidoni et al. Contemp Clin Trials 2020; Perales et al. Hisp Health Care int. 2020; Shaw et al. Ethn Health In Press; Blocker et al Kansas J Med 2020).”

19. The discussion should be abbreviated given the fact you were likely very underpowered. You may need hundreds in each arm of the study to show difference in your chosen outcomes at 12 months. It would be important to publish the natural history of change in amyloid as measured by your tracer.

>>We appreciate the reviewer’s thoughts and concerns on this. However, our approach to the results was very measured and deliberate. We have avoided making definitive declarations about effect or lack thereof. 

Reviewer #3: 

Introduction:

1. Line 65 "Higher levels of aerobic fitness are associated with age-related change in brain volume and cognition at both cross-section and over time.[3, 9-13]" - please be explicit regarding the nature of the relationship.

>>We have amended the sentence to read:

“Higher levels of aerobic fitness are associated with age-related improvements or attenuated decline in brain volume and cognition at both cross-section and over time”

2. Line 67 "In randomized controlled trials, aerobic exercise promotes brain plasticity and attenuates hippocampal atrophy while improving spatial memory.[3, 5, 14, 15]" - ref 15 reports increased hippocampal volume rather than attenuated atrophy; please amend sentence accordingly.

>>Thank you for pointing out our imprecise language. We have amended the sentence to read as follows: 

“In randomized controlled trials, aerobic exercise promotes brain plasticity, attenuate hippocampal atrophy, or even promotes hippocampal volume increase while improving spatial memory.[3, 5, 14, 15]”

3.Line 79 "...cognitively healthy adults..." - the term cognitively normal is used everywhere else in the manuscript, suggest amending for consistency.

>>We have made the suggested wording change.

4. "...and those at high genetic risk for AD.[26, 33, 34]" - ref 29 needs adding here as the reported benefits of exercise on brain amyloid are in APOE e4 carriers.

>>We have added the references as suggested. 

5. Line 91-94 "Individuals with subthreshold levels of cerebral amyloid (individuals with non-elevated amyloid PET but with quantitative measures near the threshold for being elevated) may be more likely to accumulate amyloid and have memory decline,[36] suggesting they are good candidates for

prevention studies.[37]" - to be "subthreshold" these individuals must have already accumulated some amyloid so the wording of "may be more likely to accumulate amyloid" needs amending.

>>We agree the sentence should be more precise. We have amended it to read as follows:

“Individuals with subthreshold levels of cerebral amyloid (individuals with non-elevated amyloid PET but with quantitative measures near the threshold for being elevated) may be more likely to accumulate clinically significant levels of amyloid and have memory decline,[36] suggesting they are good candidates for prevention studies.[37]“

6. Line 98 "..those with subthreshold levels of cerebral amyloid" - needs amending to '...those who are CN but with subthreshold levels of cerebral amyloid' (CN is implicit in preclinical AD in the earlier part of the sentence, but not in the subthreshold group).

>>We have changed the wording as follows to clarify the cognitive status of the “subthreshold” participants.

“Our study examined the effects of a 52-week aerobic exercise program on AD pathophysiology (amyloid burden), associated “downstream” neurodegeneration (whole brain and hippocampal volume change) and cognitive decline in cognitively normal individuals with either preclinical AD or with subthreshold levels of cerebral amyloid. “

7. Line 98-99 "We hypothesized that 52 weeks of aerobic exercise would be associated with reduced amyloid burden, reduced hippocampal atrophy,..." - "reduced amyloid burden", or a slowing of amyloid accumulation?

>>We appreciate the opportunity to provide extra precision to the hypothesis statement without being revisionist. The language in our manuscript, protocol and grant was a bit ambiguous, to be sure. We hypothesized that there would be an attenuated increase or “reduced amyloid burden” as compared to the non-aerobic exercisers. We did not expect that aerobic exercise would reduce the level of amyloid appreciably below baseline levels. We have revised the hypothesis statement to clarify this expectation. 

“We hypothesized that 52 weeks of aerobic exercise would be associated with reduced amyloid accumulation, reduced hippocampal atrophy, and improved performance on a cognitive test battery.”

Methods:

8. Line 176 "Heart Rate Reserve" - change to HRR as defined on the previous line.

>>We have amended as suggested.

9. Line 247-248 "...other covariates (age, sex, education, and PET amyloid status [elevated vs. subthreshold])..." - I am extremely interested to know if the authors controlled for baseline continuous SUVR rather than the categorical of elevated/subthreshold in any analyses undertaken that have not been reported here. As the authors rightly point out in the Discussion, amyloid accumulation is a sigmoidal curve and I do wonder if combining elevated and subthreshold may reduce the sensitivity of the analysis (particularly when combined with the test/re-test variability of FBP, and having only two amyloid scans).

Line 280 Table 1 - please change ApoE to APOE to reflect gene rather than protein.

Also, this is the first and only time that APOE is mentioned in the entire manuscript; consequently it needs defining and explaining.

On this topic, I am curious as to why APOE e4 status hasn't been included anywhere in the analysis, particularly given this statement on Line 77 "...greater amounts of self-reported physical activity (i.e., volitional behavior that is part of daily function) is associated with evidence of lower cerebral amyloid levels among cognitively healthy adults,[26-32] and those at high genetic risk for AD." where the published studies on "high genetic risk for AD" include APOE e4. Plus, we know that APOE e4 impacts amyloid accumulation, and cognitive performance (particularly in preclinical AD). Therefore, I believe this additional analysis is certainly warranted.

>>Treatment of the SUVR and APOE in our models seem to be related questions so we have addressed them together. We chose not to present these secondary analyses because we want to adhere to good CONSORT reporting practices and present clear trial outcomes. We feel it is best to report per our primary outcomes and pre-planned analyses which are consistent with our protocol and ClinicalTrials.gov. However, to be responsive to the interest of multiple reviewers and to address the possibility of differential effects of these variables of interest, we re-analyzed the data.

We tested a general linear model to examine the relationship of a continuous measure of change in SUVR. Per reviewer suggestions baseline SUVR, APOE4 status, and an interaction between exercise and E4 status were added. This model controls for baseline SUVR and includes e4 status as well as the interaction of E4 status with treatment groups to account for the imbalance of e4 status observed among groups. The p-value of 0.8385 indicated that this model was not adequate to explain the variance of change in SUVR.

Likewise, we also tested these models on the continuous measure of change in brain volume. The p-value of 0.2857 indicated that this model was not adequate to explain the variance of change in brain volume. This model with baseline whole brain volume, e4 status and related interaction is no better at explaining the variance

We used the continuous quantified values for baseline global SUVR as a covariate in our analysis. We also tested models where APOE4 was substituted for PET group (subthreshold vs. elevated) with an APOE by Treatment Arm interaction included. None of these models changed our results, and we now note this and the subsequent analysis of APOE4 in our Discussion. We have also changed ApoE to APOE throughout the manuscript. 

“As a sensitivity analysis, we also tested our models with SUVR as a covariate, and with APOE4 and APOE4 by Treatment Arm as factors. There was no appreciable change in our results in these analyses.”

10. Table 1 - no need to include % in columns when % is included in the row header. Please also define abbreviations in the Table footnotes.

>>We have amended this as suggested.

11. Table 2 - Is "Intent-to-Treat" needed twice in the Table heading? Table 3 - Is "Per-Protocol" needed twice in the Table heading?

>>We have removed the repeats.

12. Global amyloid burden / change / 0.01 (.04), missing a '0'. Please also define abbreviations in the Table footnotes.

>>We have amended as suggested.

13. Curious that the loss of brain volume in the Exercise Group is twice that of the Control Group - I wonder if this is related to, the higher % of APOE e4 carriers in the Exercise Group, where individuals are on the sigmoidal amyloid accumulation curve, or both (another reason why it could be insightful to include APOE e4 and continuous SUVR in the analysis). Please also define abbreviations in the Table footnotes.

>>Please see our response above. Inclusion of APOE4 in our models yielded no changes in our interpretation of the results.

Discussion:

14. Line 333 "... individuals who are at highest risk of amyloid accumulation." - see earlier comment regarding Line 91-94.

>>Same fix as above.

“We found no evidence that one year of aerobic exercise influences cerebral amyloid burden in a cohort of cognitively normal participants with elevated and subthreshold levels of amyloid, individuals who are at highest risk of clinically significant amyloid accumulation.”

15. Line 336-340 "Despite this, however, individuals with elevated levels of amyloid appeared resistant to aerobic exercise effects on whole brain volume, hippocampal volume, or cognitive measures. We believe these null findings support a hypothesis that the widely reported brain benefits of exercise are modest and driven mechanistically by the mitigation of non-amyloid pathologies." - this is most certainly a possibility. However, some consideration should also be given in the Discussion to the fact that it is also possible that the selected aerobic intervention wasn't effective (despite the improvements in fitness observed). Perhaps higher intensity of aerobic exercise is needed to slow amyloid accumulation?

>>We have added this a limitation. 

“Finally, it may be possible that the selected dose of exercise (duration and intensity) is insufficient or ill-suited to change amyloid accumulation. Future work should consider resistance training, or alternate intensities.”

16. Line 342 "...skewed towards fewer age-related pathologies, such as subclinical cerebrovascular disease,..." - perhaps; although it is also highly likely that your sample includes individuals with vascular amyloid deposits.

>> This is quite possible. As technologies evolve and the ability to classify individuals using the ATN (and perhaps V) framework matures, multi-domain etiological definition will be critical for clinical trials.

17. Line 362 sudden introduction of the term "Aβ" whilst "amyloid" has been used thus far - either need to define or stick with amyloid.

>>We have stuck with amyloid and amended the sentence.

18. Line 381 "Additionally, the potential benefits of aerobic exercise to influence cerebral amyloid may require a longer duration than 52 weeks." - or more intense aerobic exercise?

General Comments:

19. Please pay close attention to the comments above around factoring in APOE e4 and baseline continuous SUVR into the analysis.

>>As noted above, we are hesitant to further subdivide or add in additional covariates to planned analyses for our primary and secondary outcomes (i.e. looking at SUVR as a continuous measure). We have presented the analyses above an and briefly discussed the non-significance of APOE4 in the discussion as noted previously

20. I also wonder if the authors have considered whether there are indirect effects of the intervention on the outcome measures that relate to improvements in cardiorespiratory fitness? Specifically, there were no observed effects on amyloid, MRI and cognitive outcomes when comparing group performance from pre- to post-intervention. However, it is possible that changes in cardiorespiratory fitness from pre- to post-intervention are associated with changes in the outcome measures. It is also possible that APOE genotype may moderate these effects.

>>This is not only possible but likely. However, as in our prior work, we have elected to largely report our findings following CONSORT guidance, saving secondary analyses that were not pre-planned for follow up reports. Vidoni et al PLOS One 2015; Billinger et al. JAPA 2017). It does not appear that APOE4 effects are sufficiently strong to impact our results. However, we do note in the limitations that block randomization based on APOE4 may be a valuable strategy in the future.

For a later date, it may well also be valuable to assess the impact of the intervention of blood levels of amyloid species using the new generation of ultra-sensitive assays that appear to be extremely promising.

We agree and are very excited about the prospect of these assays. Imaging is valuable but we recognize its limitations.

Reviewer #4: 

1. The authors should give details of how the randomisation schedule was produced, eg. whether blocking or stratification were used etc.

>>We have provided additional information in the Allocation section.

“A study statistician constructed an allocation schedule that was applied by study staff after baseline testing was completed. The study statistician used random number generator to generate blocks of nine in a 2:1 ratio to protect against imbalance if recruitment fell short Participants were prospectively assigned to treatment versus control from this schedule using REDCap’s randomization module which restricts access and viewing once uploaded.”

2. I think that the first sentence of the statistical analysis section should read ‘…including means, standard deviations and…’

>>We have amended this as requested.

3. Statistical analysis, second sentence: two-sample t-tests and paired t-tests are different things. What the authors should have used here are two-sample t-tests comparing the change from baseline between the two groups. I am not sure which they have actually used.

>>We have corrected the error. We used two-sample t-tests of change scores.

4. The intention to treat and per protocol populations should be defined in the methods rather than in the results.

>>We define the criterion for inclusion in ITT vs PP in the Adherence and Safety Section of the Methods. Because we do not present overall enrollment in the Methods, we feel it is best to keep the specific size of these sets in the Results where we discuss the overall enrollment.

5. The results from Tables 2 and 3 are reported in the text only in terms of their p-values. It would be helpful for the authors to discuss the clinical significance of the differences, even where there is a lack of statistical significance. Eg. Whole brain volume has a p-value >0.05 but there does seem to have been more of a decrease in the exercise group. Is the magnitude of the difference clinically significant? (I am a statistician and I genuinely do not know if the drops in whole brain volume and the magnitude of the difference between the groups is clinically relevant).

>>This is a really important question, but also one that is difficult to answer. Fjell et al (2009 J Neurosci) reported 1-year rates of average volume change in cognitively normal older adults from 0.2 to 0.8% depending on region. Specific hippocampal volume loss average 0.84%. Atrophy is also dependent on age. This makes any definitive statement on our results difficult. However, our atrophy rates of 0.2 to 0.5% globally and 0.9-1.0% don’t appear to be particularly outside the normal range. Fjell also reported global atrophy rates of atrophy in people with dementia to be 0.2-3.8% depending on region while hippocampal atrophy averaged 3.75% in people with dementia. Thus, our atrophy rates fall well below that which might be associated with clinical symptoms.

We are hesitant to make any definitive statement on clinical significance, or lack thereof, of our data. However, we have added the following sentence to our Discussion.

“Our observed atrophy rates are consistent with those previously reported in cognitively normal older adults (Fjell et all 2009).”

6. Table 2: number formats should be the same within outcomes, eg. Baseline global amyloid should be 1.20 rather than 1.2 and the SE of the change should be 0.04, not .04.

>>Amended as suggested.

---

## [Decision Letter · Decision Letter 1]

8 Dec 2020

PONE-D-20-24208R1

Effect of Aerobic Exercise on Amyloid Accumulation in Preclinical Alzheimer’s: A 1-Year Randomized Controlled Trial

PLOS ONE

Dear Dr. Burns,

Thank you for submitting your manuscript to PLOS ONE. The revised version is acceptable but has one small error still in need of correcting. Please submit a revised version of the manuscript that addresses this.

We look forward to receiving your revised manuscript.

Kind regards,

Ashley I Bush, MD PhD

Academic Editor

PLOS ONE

Reviewers' comments:

Reviewer's Responses to Questions

**Comments to the Author**

1. If the authors have adequately addressed your comments raised in a previous round of review and you feel that this manuscript is now acceptable for publication, you may indicate that here to bypass the “Comments to the Author” section, enter your conflict of interest statement in the “Confidential to Editor” section, and submit your "Accept" recommendation.

Reviewer #1: All comments have been addressed

Reviewer #2: All comments have been addressed

Reviewer #3: All comments have been addressed

Reviewer #4: (No Response)

2. Is the manuscript technically sound, and do the data support the conclusions?

Reviewer #1: Yes

Reviewer #2: (No Response)

Reviewer #3: (No Response)

Reviewer #4: Yes

3. Has the statistical analysis been performed appropriately and rigorously? 

Reviewer #1: Yes

Reviewer #2: (No Response)

Reviewer #3: (No Response)

Reviewer #4: Yes

4. Have the authors made all data underlying the findings in their manuscript fully available?

Reviewer #1: Yes

Reviewer #2: (No Response)

Reviewer #3: (No Response)

Reviewer #4: Yes

5. Is the manuscript presented in an intelligible fashion and written in standard English?

Reviewer #1: Yes

Reviewer #2: (No Response)

Reviewer #3: (No Response)

Reviewer #4: Yes

6. Review Comments to the Author

Reviewer #1: (No Response)

Reviewer #2: (No Response)

Reviewer #3: (No Response)

Reviewer #4: I commend the authors for thoroughly addressing my comments. However, one has been missed:

"I think that the first sentence of the statistical analysis section should read ‘…including means, standard deviations and…’ "

- The word 'deviations' is still missing.

7. PLOS authors have the option to publish the peer review history of their article (what does this mean?). If published, this will include your full peer review and any attached files.

Reviewer #1: No

Reviewer #2: **Yes: **Amy Brodtmann

Reviewer #3: No

Reviewer #4: **Yes: **Sarah Barry

---

## [Author Response · Author response to Decision Letter 1]

15 Dec 2020

Dr. Bush,

Academic Editor, PLOS One

We appreciate the final reviews of our manuscript PONE-D-20-24208R1, “Effect of Aerobic Exercise on Amyloid Accumulation in Preclinical Alzheimer’s: A 1-Year Randomized Controlled Trial”. We have made the final requested correction

In the analysis section our sentence now reads:

“Descriptive statistics were generated, including means, standard deviations and ranges for continuous measures, and frequencies and relative frequencies for categorical measures.”

We have provided a “clean” copy of our revised manuscript. We thank the reviewers for helping us to improve the manuscript and look forward to positive reception.

Sincerely,

Jeffrey M. Burns, MD, MS

University of Kansas Medical Center.

---

## [Editor Report · Decision Letter 2]

18 Dec 2020

Effect of Aerobic Exercise on Amyloid Accumulation in Preclinical Alzheimer’s: A 1-Year Randomized Controlled Trial

PONE-D-20-24208R2

Dear Dr. Burns,

We’re pleased to inform you that your manuscript has been judged scientifically suitable for publication and will be formally accepted for publication once it meets all outstanding technical requirements.

Kind regards,

Ashley I Bush, MD PhD

Academic Editor

PLOS ONE
---

## [Editor Report · Acceptance letter]

22 Dec 2020

PONE-D-20-24208R2 

Effect of Aerobic Exercise on Amyloid Accumulation in Preclinical Alzheimer’s: A 1-Year Randomized Controlled Trial 

Dear Dr. Burns:

I'm pleased to inform you that your manuscript has been deemed suitable for publication in PLOS ONE. Congratulations! Your manuscript is now with our production department. 

Kind regards, 

on behalf of

Dr. Ashley I Bush 

Academic Editor

PLOS ONE